# Common Running Musculoskeletal Injuries and Associated Factors among Recreational Gorge Marathon Runners: An Investigation from 2013 to 2018 Taroko Gorge Marathons

**DOI:** 10.3390/ijerph17218101

**Published:** 2020-11-03

**Authors:** Chia-Li Hsu, Chich-Haung Yang, Jen-Hung Wang, Chung-Chao Liang

**Affiliations:** 1Department of Physical Medicine and Rehabilitation, Hualien Tzu Chi Hospital, Buddhist Tzu Chi Medical Foundation, Hualien 97004, Taiwan; 108331101@gms.tcu.edu.tw (C.-L.H.); STONE@tzuchi.com.tw (C.-C.L.); 2Department of Physical Therapy, College of Medicine, Tzu Chi University, Hualien 97004, Taiwan; 3Sports Medicine Center, Hualien Tzu Chi Hospital, Buddhist Tzu Chi Medical Foundation, Hualien 97004, Taiwan; 4Department of Medical Research, Hualien Tzu Chi Hospital, Buddhist Tzu Chi Medical Foundation, Hualien 97004, Taiwan; jenhungwang2011@gmail.com

**Keywords:** gorge marathon, running injury, injury incidence, risk factor

## Abstract

Many studies exist on the incidence and related risk factors of running injuries, such as those obtained during marathons. However, in gorge-terrain marathons, an insufficient number of reports exist in the relevant literature. Therefore, this study aimed to explore the incidence of musculoskeletal injuries occurring in participants in the 2013 to 2018 Taroko Gorge Marathons in Taiwan and the distribution of running injuries and related influencing factors. A total of 718 runners who entered the physiotherapy station presented with records of treatment and injuries and filled out a running-related injury and self-training questionnaire for further statistical analysis. The association between risk factors and injury were evaluated by logistic regression. The injured areas on the lower extremities after the gorge marathon were as follows: 28% in the knees, 20% in the posterior calves, 13% in the thighs, 10% in the ankles, and 8% in the feet. The analysis of injury-related risk factors showed that male athletes demonstrated a higher risk of thigh injury than female athletes (OR = 2.42, *p* = 0.002). Underweight runners exhibited a higher risk of thigh injury (OR = 3.35, *p* = 0.006). We conclude that in the gorge marathon the rates of knee, calf, thigh, and foot injuries are significantly increased. Medical professionals, coaches, and runners may use the findings of this study to reduce the potential risk of running injuries in marathons.

## 1. Introduction

Running is the sport of choice of many individuals and the general public because of its health advantages in addition to its convenience and low cost. Running is a type of popular, but not necessarily extremely skillful, exercise that is not limited by body shape, age, or sex. Since the 1970s, running has been increasing in popularity globally. Previous studies reported a one-year prevalence of 54.8% and incidence from 19.4% to 79.3% of musculoskeletal injuries during long-distance running [1,2].

Most injuries related to running occur in the lower extremities [2,3,4,5,6]. The most common anatomical site for running injuries is the knee [1,6,7,8]. Although some injuries are traumatic, most are caused by overuse. The definition of “injury” differs in the literature. However, the most common definition of running-related injury (RRI) is as follows: musculoskeletal disease caused by running, which results in limited running speed, distance, duration, or frequency for at least one week [8]. The cause of RRI is usually related to repeated musculoskeletal trauma and can be attributed to various risk factors, for example, the impact of runners’ personal characteristics (anatomical and biomechanical factors), training errors (e.g., training volume, weekly distance), running experience [9,10], and previous RRIs [11,12]. However, conflicting evidence exists for other risk factors, such as age and sex [3,10,13], training distance [4,5,7,10], running experience [14], body mass index (BMI) [15], and orthosis use [5,16]. Previous studies that attempted to reduce running injuries focused on preconditioning, warmup/cooldown, shoe type modification, and a directed, graduated training program [17,18]. Therefore, many studies reported different theoretical concepts.

The Taroko Marathon takes place in the Taroko National Park, Hualien, Taiwan. It has been held every year since 2000. Its magnificent and unique canyon landscape attracts runners from all over the world, with more than 10,000 runners participating each year. The terrain features frequent changes in altitude. It is generally believed that running on rough terrain and slopes may be related to the increased occurrence of musculoskeletal injuries [19,20,21,22,23]. Many studies about running on mountain roads and on uphill and downhill terrains have shown that this training causes muscle fatigue and injury [19,24]. When running uphill and downhill, the increase in the angles of the lower limb joints and the increase in impact are factors that may cause musculoskeletal injuries [20,21,22].

Therefore, we hypothesized that the musculoskeletal injury distribution in the lower limbs may significantly increase, and associated factors could be significantly related to the injuries, in gorge marathon runners compared to those of marathon runners generally, based on the results of a self-report questionnaire. This study aimed to explore the incidence of musculoskeletal injuries occurring in gorge marathon runners and examine the distribution of running injuries and related influencing factors.

## 2. Materials and Methods

This was a cross-sectional and retrospective study. We recorded the management of injuries that the participants of the 2013–2018 Taroko Marathons received. Because of injury or discomfort, these recreational runners visited the physical therapy station for further consultations after the initial evaluation. Once these recreational runners entered the physical therapy station, a questionnaire to record the basic characteristics, injury area, and physical therapy intervention was completed by the physical therapists after interviewing the runners. The runners also needed to fill out a questionnaire about their injury and self-training. 

After obtaining consent from the organizer, we obtained the recorded data, which was filled in anonymously, and no personal identification information was present, except for the player number. Musculoskeletal injuries and related factors of the marathon participants were analyzed and were discussed later in the paper. The protocol was approved by the Research Ethics Committee of Hualien Tzu Chi Hospital, Buddhist Tzu Chi Medical Foundation (IRB l08-226-B).

### 2.1. Subjects

From 2013 to 2018, we provided a questionnaire to all runners who were enrolled in the Taroko Gorge Marathon and sustained an injury during the race for further investigation about their injury and therapy records. Due to injury or discomfort after the race, these runners received physical therapy. The inclusion criteria for the recreational marathon runners were as follows: participants in the Taroko Marathon, adults aged > 18 years, voluntarily came to the physical therapy tent to seek treatment or consultation after the marathon, and willing to fill in the questionnaire. The exclusion criteria were as follows: no special requirements were present; all individuals who visited the physical therapy station for further treatment or consultation filled out the sports injury and therapy questionnaire. In this study, a total of 718 questionnaires were collected from 2013 to 2018. After the questionnaires were collected and reorganized, 9 participants with incomplete questionnaires were excluded. Finally, a total of 709 questionnaires were included in the analysis.

### 2.2. Characteristics of Taroko Gorge Marathon

The Taroko Marathon takes place in Taroko National Park, Hualien, Taiwan. This unique route features an altitude increase and then decrease of 500 m. The first 7 km are flat, followed by a continuous uphill route for 18 km, and then a downhill route for 18 km. The uphill slope rises from 0 m at 7 km to 500 m at 25 km, the slope of this Taroko-Tianxiang section ranges from 0% to 10%, and the average slope is 5%; the downhill route decreases from 500 m at 25 km to 0 m in the last kilometer of the race. The route travels through several tunnels throughout the race.

### 2.3. Instrument

The questionnaire in this study was based on a questionnaire used in previous studies [1,4,7] related to musculoskeletal injuries of the lower extremities in marathon runners. The main contents of questionnaire were employed from the questionnaire designed by Chang et al. [4] (see the Appendix A). Before our questionnaire was officially used in the study, some experts in clinical medicine and sports science reviewed the questionnaire and provided comments and suggestion on controversial and/or unclear questions. Finally, we made adjustments to the questionnaire based on a pilot experiment. The questionnaire contained five parts: basic information, marathon experience, training methods, running-related symptoms, and physical therapy treatment records. Basic information included the following: age, sex, height, weight, and participation in the project. Running experience included participation in the marathon exercise experience, number of weekly runs (running frequency), and distance and time of each run. The training methods included whether special training was present, frequency, and distance of training. Running-related symptoms included investigation of injury and pain areas. Physical therapy treatment records and satisfaction included protection and treatment sites, methods, and improvement before and after. The questionnaire included 10 items. Except for questions on age, height, and weight, which were open-ended questions, all were categorical response questions.

Before the questionnaire was officially used, experts were asked to review the questionnaire, delete controversial and unclear questions, and make small adjustments to the questionnaire based on the results.

### 2.4. Statistical Analysis

After collection of the questionnaires, they were classified and counted. Microsoft Excel was used for data sorting, and then the Statistical Package for Social Science 17 (SPSS Inc., Chicago, IL, USA) program software was used for data analysis and statistics. The descriptive statistics were used to present the distribution of road runners’ personal data, running experience, training methods, and injuries.

The basic characteristics of runners who received physical therapy were classified and compared using the chi-square test. Continuous variables, such as age and body weight, were analyzed using the independent t-test. The chi-square test was used to analyze previous injuries and injured parts of runners who received physical therapy. The studied risk factors related to the musculoskeletal injury were age, sex, BMI, education level, type of competition, and previous injuries. The association between risk factors and musculoskeletal injury were evaluated by logistic regression. Odds ratio was adopted to quantify the effect.

## 3. Results

### 3.1. Basic Characteristics

For the data collection from 2013 to 2018, 718 marathon runners who voluntarily received physical therapy were included for further analysis of the questionnaire. However, nine participants had incomplete basic data and were excluded from the data analysis. A total of 709 runners were included. In the comparison between men and women, some important findings were present as shown in the following: (1) Male runners receiving physical therapy exhibited significantly higher BMI and overweight and obesity proportions than female runners. (2) A comparison of academic qualifications showed that the proportion of male runners with a master’s degree or above was significantly higher than that of female runners. (3) The comparison of running types demonstrated that the proportion of female runners who participated in the gorge marathon was significantly lower than that of male runners; female runner’s marathon experience and weekly running frequency, distance, or duration were also less than that of male runners. 

Approximately 76% of runners presented with previous sports injuries, while 97% of runners reported injury after attending this type of the gorge race. Approximately 68% of runners attributed their discomfort or injury to a gorge-type marathon (Table 1).

### 3.2. Previous Injured Sites of the Body before Gorge Marathon

Based on the statistical analysis of the injured sites of the runners who received physical therapy, we found that the proportions of previously injured sites in these runners were 36% in the knee, 26% in the ankle, 15% in the back, 14% in the calf, and 10% in the foot. No significant difference was found in the proportion of male and female runners who were injured in various sites as shown in Figure 1.

### 3.3. Proportion of Injured Sites of the Body after the Gorge Marathon

We found that the runners who received physical therapy presented with injuries in the following sites: 38% in the knee, 20% in the thigh, 19% in the ankle, 32% in the calf, and 16% in the foot. The proportion of calf injury (men, 34%; women, 28%; *p* = 0.095) and thigh injury (male, 25%; female, 13%; *p* < 0.001) in male runners was significantly higher than that of female runners. Female runners presented with a significantly higher percentage of hip injuries than male runners (men, 4%; women, 10%; *p* = 0.001) (Figure 2).

### 3.4. Comparing the Proportions of Injured Sites before the Gorge Marathon

By comparing the proportion of previously injured sites with injuries attributed to the current marathon event, we found that the incidence of injury in the calf (+18%), thighs (+12%), and feet (+6%) was significantly higher than the background value. The definition was set as the newly injured calf as the injured sites on this side—previous injured sites, which is the formula as shown below, for example, injury in the calf muscle:New calf injury = Currently calf injury − Formerly calf injury

### 3.5. Influenced Sites of the Body after the Gorge Marathon

Based on the statistical analysis on the perceived effects of the gorge-typed marathon, we found that the proportion of injury sites the runners felt were attributable to the gorge-type marathon were 28% in the knee, 20% in the calf, 13% in the thigh, 10% in the ankle, and 8% in the foot. Male runners showed a significantly higher impact of the gorge-type marathon for thigh injures than did female runners (men, 15%; women, 9%; *p* = 0.014) (Figure 3).

### 3.6. Factors Associated with Injury

The risk factors related to injury that were attributed to the participation in the gorge-type marathon included age, sex, BMI, education level, and type of competition. The statistical analysis of previous injuries showed categorically the following (Table 2):

1. The analysis of the risk factors related to calf injury showed that runners with previous calf injury exhibited significantly higher risk than runners who presented with no previous calf injury (OR = 4.02, *p* < 0.001).

2. The analysis of risk factors related to thigh injury showed that male runners demonstrated a higher risk of injury than female runners (OR = 2.42, *p* = 0.002). Underweight runners demonstrated a higher risk of thigh injury (OR = 3.35, *p* = 0.006). Runners with previous thigh injury demonstrated significantly higher risk than runners who presented with no previous thigh injury (OR = 9.66, *p* < 0.001). 

3. The analysis of risk factors related to knee injury showed that the older the individual, the lower the risk of knee injury (OR = 0.98, *p* = 0.019). Underweight runners demonstrated a significantly lower risk of knee injury than runners with normal weight (OR = 0.32, *p* = 0.017). Runners with previous knee injury demonstrated significantly higher risk than runners who presented with no knee injury (OR = 5.65, *p* < 0.001).

## 4. Discussion

This retrospective self-report study on injured Taroko Marathon runners showed that approximately 68% presented with injuries in the knee, calf, and thigh. The study also found a difference in injuries between men and women. Male runners exhibited more marathon experience and longer weekly running times and distance than those of female runners, so the incidence of injuries also increased. Generally, an increase in BMI seemed to increase the risk of RRI. 

### 4.1. Running Injury Incidence and Injury Location

This study analyzed the characteristics of running injuries and training methods among 709 participants of the Taroko Marathon from 2013 to 2018. The results found that, consistent with previous literature, RRIs are mainly lower limb problems. During the marathon, 97% of runners presented with injuries to their lower limbs, and 68% of runners thought it was influenced by the Taroko Marathon, in which knee pain was the most common running-related problem. Steinacker et al. (2001) proposed that the incidence of lower limb injuries of marathon runners was 9–50%. They also discovered that the common injured site was the knee joint, accounting for 43.9% of injuries, followed by the Achilles tendon [25]. Middelkoop et al. (2008) proposed that the incidence of lower extremity injuries of marathon runners accounted for 79.6% of injuries, and the common injured site was the knee joint, accounting for 26.7% of injuries, followed by the posterior side of the lower leg (14.4%) and the anterior side of the thigh (13.8%) [1]. Chang et al. (2011) conducted a study on the 2005 ING Taipei International Marathon that included three categories: marathon, half-marathon, and 10 km race. It was found that 196 of 893 (32.5%) marathon runners presented with knee injuries. The incidence of lower limb injuries accounted for the highest proportion of the injuries, at approximately 48% [4]. Ellapen et al. (2013) showed lower limb injuries in 180 half-marathon runners, of which the knee joint was the most commonly injured site (26%), followed by the tibia/fibula (22%) and lower back and hip joint (16%) [26]. Mayooran et al. (2019) reported that the incidence of lower extremity injuries in marathon runners was 55.85%, and the most commonly injured site was the knee joint, accounting for 22.38% of injuries [27]. Interestingly, in a recent case study, Gajda et al. (2020) tested a 36-year-old male ultramarathon runner before and after winning a 24 h ultramarathon. After running for 12 h (about 130 km), the athlete experienced pain in his right knee for the first time. This pain intensified and affected his performance. One day after running, magnetic resonance imaging (MRI) of the right knee showed overload and degeneration of the right lower limb and related specific characteristics [8]. A previous systematic review (van Gent et al., 2007) reported that the incidence of running injuries in the lower extremities in long-distance runners varied from 19.4% to 92.4%. The most common site of lower extremity running injuries was the knee [2]. The overall proportion of injury by specific pathology was reported in 11 studies. The knee (28%), ankle-foot (26%), and shank (16%) accounted for the highest proportion of injury in male and female runners [28].

Based to the occurrence and distribution of the abovementioned injuries, it is clear that sports injuries are also slightly different due to the different bone and muscle conditions of male and female runners. This study found that the proportion of calf (men, 34%; women, 28%) and thigh (men, 25%; women, 13%) injury in male runners was higher than that of female runners. Female runners exhibited a higher percentage of hip injuries than did male runners (men, 4%; women, 10%). Ellapen et al. (2013) found that the main injured sites of men were the knee (27%), tibia/fibula (20.2%), and thigh (15.7%), while the main injured sites of women were the knee (25.9%), tibia/fibula (23.32%), and lower back/hip (16.4%) [26]. Their results were confirmed by the results of this study. In female runners, the incidence of hip injuries is high. 

The Taroko Marathon is a gorge-type race and, therefore, exhibits uphill and downhill slopes, suggesting that it may increase the incidence of injury. Vernillo et al. (2017) showed that running uphill is characterized by a higher stepping frequency, greater internal mechanical work, shorter duration of the swing/air phase, and greater duty cycle, while running downhill is characterized by a longer flight time, lower step frequency, and reduced duty cycle. When running uphill, the lower limb muscles perform a higher net mechanical work to increase potential energy. In downhill running, as the running gradient increases, the demand for work also increases, which is met by increasing the power output of all joints, especially the hips [24]. Giovanelli et al. (2016) found that the greater maximum mechanical power of the lower limbs is related to changes in running mechanics caused by fatigue. The uphill marathon requires training of lower limb strength, which may improve running performance [20]. Carmona et al. (2015) reported that muscle damage after prolonged mountain running and an increase in the concentration of slow myosin serum after mountain ultramarathons could be indirect evidence of slow-(type I) fiber-specific sarcomere disruptions [21]. Malliaropulos et al. (2015) found that ultra-trail-runners and higher-level runners present with more musculoskeletal injuries than lower-level runners, with symptoms most commonly developing in the lower back, hip joint, and plantar surface of the foot. Experienced runners (>6 years) are at greater risk of developing injuries, especially in the lower back, tibia, and foot plantar surface [29]. Telhan et al. (2010) analyzed lower limb kinetics during moderately sloped running to show that, when downhill running, the knee absorption and hip strength may increase. Moreover, hip absorption during downhill running when runners straightened up early in their gait cycle increased [23]. Therefore, this study verified the incidence running-related injuries to the lower limbs and their distribution location, which confirms the research and hypotheses of previous scholars.

### 4.2. Risk Factors for Running Injuries 

#### 4.2.1. Sex

The results of previous studies using sex grouping analysis found that male runners demonstrated a higher injury rate than that of female runners in the analysis of risk factors. Vitez et al. (2017) explored the incidence of marathon runners’ injuries and risk factors. The study included a total of 697 runners, with 240 women (49%) and 357 men (51%). The statistical results indicated that men accounted for a relatively large proportion of both smaller (55%) and larger injuries (57%) [30]. van Poppel et al. (2015) explored the incidence and risk factors of running injuries of half-marathon and marathon runners. Male participants accounted for 67.4% of the total participants. Among all runners, 67% of half-marathon participants were male, while 68.7% of marathon participants were male. According to statistical results, the odds ratio for men injured in the half-marathon was 1.44, which is greater than 1 and so a confirmed risk factor [31]. In a 2013 study, Rasmussen et al. explored if the risk of running injury would decrease with the increase in weekly running amount before the marathon race. They showed that 52 of 535 men and 16 of 127 women were injured. When women were compared to men, the relative risk was 0.77. No association between sex and RRI was found in the study. In contrast, the literature pointed out that more male runners exist than female runners. Men present with more injuries than women [32]. This study found that the injury rate of male runners was higher, with 453 male runners (63%) of 709 runners suffering injuries.

#### 4.2.2. BMI

In terms of BMI, based on the study of Vitez et al. (2017), the average BMI of professional runners was 23.1 ± 2.6 kg/m^2^. The results showed that runners with a BMI > 25 kg/m^2^ exhibit a higher risk of injury [30]. van Poppel et al. (2015) showed that a BMI > 26 kg/m^2^ in half-marathon runners is a risk factor of running injuries [31]. Rasmussen et al. (2013) mainly explored if different weekly running distances results in different risk factors for running injuries for obese and non-obese individuals. They divided runners into two groups based on BMI difference, >30 kg/m^2^ and <30 kg/m^2^. Comparing runners after 10 km and 20 km runs, the results showed that a statistically significant difference was found in BMI > 30 kg/m^2^ [32]. Nielsen et al. (2014) revealed that the BMI of professional runners was 23.0 ± 2.3 kg/m^2^. The average BMI of runners with a high risk of injury was 23.1 ± 2.2 kg/m^2^. Comprehensive literature results show that runners with higher BMI (>25 kg/m^2^) exhibit a higher risk of injury [33]. 

In this study, the average BMI of runners was 22.62 ± 3.08 kg/m^2^. The average BMI of male runners was 23.59 ± 3.06 kg/m^2^, and the average BMI of female runners was 20.81 ± 2.17 kg/m^2^. Interestingly, in our study, underweight runners had a lower risk of knee injury after the gorge marathon, which may imply a reduced biomechanical factor, such as body weight impact, on the knee during a gorge marathon. In contrast, we did not find any significant risk factor in calf, thigh and knee injuries for overweight and obese runners. Although no higher incidence of injury was found for higher BMIs (>25 kg/m^2^), a statistically significant difference was found between men and women, which is similar to the results of Rasmussen et al. (2013). In addition, Nielsen et al. (2013) suggested that an increase in BMI would increase the load on lower limb exercise, which would increase the risk of running injury [33]. Therefore, if the overweight person reduces running, they are protected from sports injuries because it reduces the stimulation of training and thus decreases the risk of injury [32,33].

More importantly, overweight and obese runners often exhibit a larger load per stride due to weight increase, which indicates that fewer repetitions of the stride are required to accumulate the same cumulative load as a normal-weight runner [34]. Therefore, overweight and obese runners may be at greater injury risk earlier in running than normal-weight runners. Researchers recommend that to minimize these two mechanisms runners should reduce the number of strides/distance accordingly to adjust the cumulative load to a level that is unlikely to cause damage.

#### 4.2.3. Running Training Factors Include Running Experience, Training Frequency, and Running Distance

Our finding showed that marathon experience (years) (men, 2.22 ± 1.8 years; women, 1.85 ± 1.65 years), weekly running frequency (times/week) (men, 2.73 ± 2.01 times/week; women, 2.33 ± 1.71 times/week), distance (km) (men, 8.46 ± 11.39 km; women, 6.33 ± 3.3 km), and duration (min) (men, 50.02 ± 25.45 min; women, 43.17 ± 23.46 min) of female runners are also lower than those of male runners. A previous systematic review reported that women exhibit lower risk of sustaining RRIs than men. A history of injury, running experience of 0–2 years, restarting running, weekly running distance of 20–29 miles, and running distance > 40 miles per week were associated with a greater risk of RRI in men than in women [35]. Meanwhile, other studies reported that male runners exhibit a greater training distance per week, and a history of injury is a risk factor of lower extremity running injuries [2].

For running experience, Vitez et al. (2017) showed that the probability of running injury with running experience of 4 to 10 years is relatively high [30]. van Poppel et al. (2015) divided running experience into three groups. The results showed significant differences, with runners demonstrating higher running experience from 0 to 4 years in half-marathons, with an odds ratio of 1.77. An analysis was conducted on half-marathon runners, and the results showed that the odds ratio for runners with <5 years of running experience was 1.87. Therefore, the probability of injury for runners <5 years is higher [31]. The study of Chang et al. (2012) mainly explored the distribution of lower limb running injuries and its influencing factors. Most runners of this study had 1–5 years of running experience. The incidence of lower limb injuries of half-marathon runners was 42.2%, while the incidence of lower limb injuries of marathon runners was 48% [4].

Comprehensive literature results showed that more running injuries also occurred in runners with <5 years of running experience. More importantly, we also showed that the average running experience for injured gorge marathoners was between 1 and 3 years, which was consistent with the previous literature. Middelkoop et al. (2008) suggested that the reason that experienced runners are less likely to be injured is because their musculoskeletal system has adapted to running. Consequently, they demonstrate the ability to interpret their own body signals, so they will be properly trained before an injury occurs [7].

For the frequency of training, van Poppel et al. (2015) reported that for marathon runners, a training frequency of >5 times per week will lead to running injuries. In this study, 2.8% of half-marathon runners and 3.3% of marathon runners trained >5 times a week. They provided two possible explanations: The first explanation is that it seems that this study excluded competitive athletes, who will do more intensive training than will amateur runners. The second explanation is that, although half-marathon and marathon runners demonstrate a lower training frequency, the training distance is longer [31]. A study by Chang et al. (2012) found that runners who ran >5 times a week presented with more sprained feet (*p* = 0.009), and runners who ran 2–5 times a week presented with more ankle pain (*p* = 0.026) [4]. Specifically, the two studies from Middelkoop et al. (2008) and Mayooran et al. (2019) reported that weekly running frequency showed no significant differences in long- and short-distance runners [7,27]. Comprehensive literature results showed that running injuries occur more often with >5 training sessions per week. However, the results in our study can only show that runners train approximately 1–4 times per week with a greater injury rate in men.

For the training distance, Vitez et al.’s (2017) research results showed that when runners train for 21–30 km and 31–50 km a week, the incidence of running injuries is higher, and the main injury site is the foot [30]. van Poppel et al. (2015) showed that the incidence of injury among marathon runners whose training distance is <20 km per week is higher [31]. Rasmussen et al. (2013) reported that runners with a weekly training distance of 0–30 km presented with a significantly increased risk of injury (134%) compared with runners with a weekly training distance of 30–60 km [32]. Chang et al. (2012) found that when runners’ training distance per week is <20 km, the incidence of lower limb injuries of half-marathon runners was 42.2%, and the incidence of lower limb injuries of marathon runners was 48% [4]. 

The results of comprehensive literature reviews [4,30,31,32] indicate that the weekly training distance of runners from 0 to 30 km may increase the risk of injury. The results in our study showed that the injured runners’ running distance per week was <20 km. This is also consistent with previous research results that showed that the shorter the training distances per week, the higher the risk of running injuries.

#### 4.2.4. Limitations

The limitations of this study are that it was conducted in a cross-sectional research design and we collected the data using a self-report questionnaire based on voluntary participation from the runners who sought the assistance or consultation for their injury condition at a physical therapy station. Therefore, there are some limitations to the interpretation of the results of this study. First, there was some recall bias as the participants had to fill in their questionnaires in a hectic manner during the treatment period. Second, due to the limitation of the research design and instrumentation, the cause–effect relationship between musculoskeletal injuries and associated factors may not be determined from our findings. Further studies could enhance the process of diagnosis, physical examination before the enrollment, and completion of the questionnaire.

## 5. Conclusions

In summary, of the 709 runners in the final study sample, 506 (75.7%) reported injuries or pain in the lower limbs previously associated with running. The study showed that the injury site in runners who received physical therapy was mainly the knee, with approximately 38% of all injuries. The percentage of male runners with calf and thigh injuries was significantly higher than that of female runners, while females exhibited a significantly higher percentage of hip injuries than did men. When comparing injuries with a previous injury site, we found that the proportion of calf, thigh, and foot injuries significantly increased. The results show that the gorge-typed marathon may demonstrate a greater impact on these parts, leading to running injuries. In terms of sex differences, we found that men’s marathon experience and self-training intensity are also risk factors for injury. Medical professionals, coaches, and runners may use the findings of this study to reduce the potential risk of running injuries in marathons.

## Figures and Tables

**Figure 1 ijerph-17-08101-f001:**
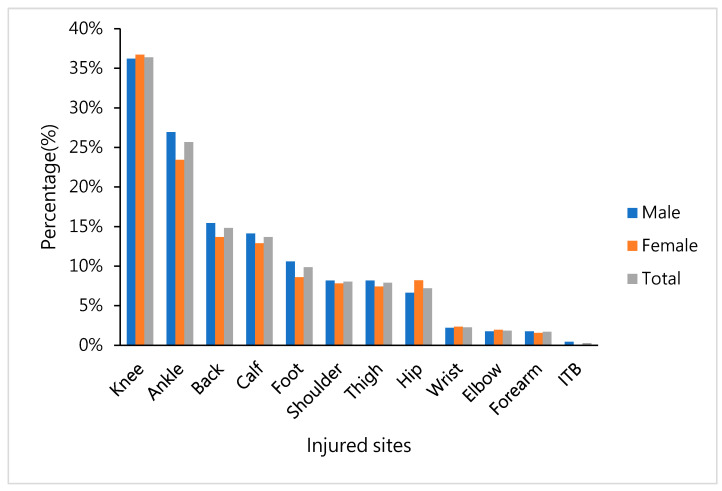
Percentage of self-reported previously injured sites in the body before the gorge marathon, shown for men, women, and total percentage.

**Figure 2 ijerph-17-08101-f002:**
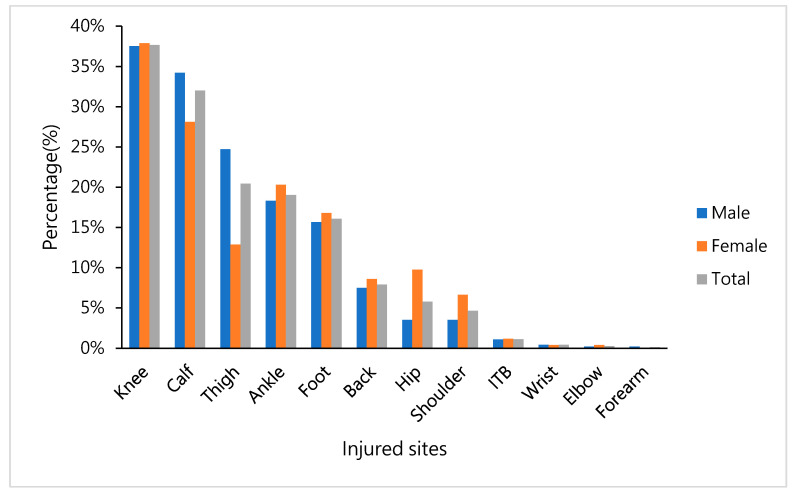
Percentage of self-reported injury sites in the body after a gorge marathon, shown for men, women, and total percentage.

**Figure 3 ijerph-17-08101-f003:**
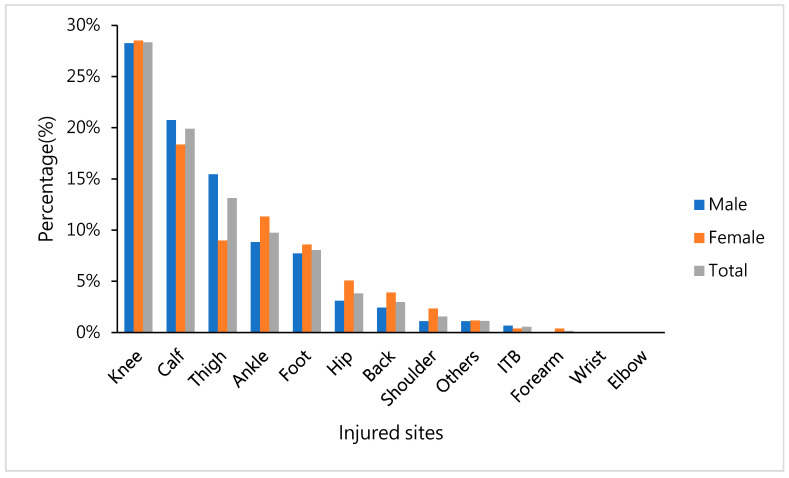
Percentage of self-reported influence on the injury sites in the body after the gorge marathon, shown for men, women, and total percentage.

**Table 1 ijerph-17-08101-t001:** Demographic data for the runners who participated in this study from 2013 to 2018 (n = 709).

Characteristics	Male	Female	Total	*p* Value
N	453	256	709	
Year	-	-	-	0.517
2013	74 (16.3%)	40 (15.6%)	114 (16.1%)	
2014	67 (14.8%)	39 (15.2%)	106 (15.0%)	
2015	83 (18.3%)	43 (16.8%)	126 (17.8%)	
2016	88 (19.4%)	38 (14.8%)	126 (17.8%)	
2017	82 (18.1%)	55 (21.5%)	137 (19.3%)	
2018	59 (13.0%)	41(16.0%)	100 (14.1%)	
Age	34.56 ± 9.98	33.09 ± 9.71	34.03 ± 9.91	0.063
BMI	23.59 ± 3.06	20.81 ± 2.17	22.62 ± 3.08	<0.001*
BMI group	-	-	-	<0.001 *
Normal	259 (58.5%)	187 (79.2%)	446 (65.7%)	
Underweight	9 (2.0%)	29 (12.3%)	38 (5.6%)	
Overweight	124 (28.0%)	18 (7.6%)	142 (20.9%)	
Obese	51 (11.5%)	2 (0.8%)	53 (7.8%)	
Education	-	-	-	0.001 *
High school degree or below	35 (8.0%)	19 (7.7%)	54 (7.9%)	
College degree	224 (51.5%)	161 (65.4%)	385 (56.5%)	
Master’s degree or above	176 (40.5%)	66 (26.8%)	242 (35.5%)	
Type	-	-	-	<0.001 *
Mini-marathon (5K)	10 (2.5%)	22 (9.8%)	32 (5.1%)	
Half-marathon	325 (79.9%)	191 (85.3%)	516 (81.8%)	
Marathon	72 (17.7%)	11 (4.9%)	83 (13.2%)	
Experience of running	-	-	-	
Mountain road marathon (%)	194 (46.1%)	82 (34.6%)	276 (41.9%)	0.004 *
Self-training (%)	87 (22.5%)	40 (18.5%)	127 (21.1%)	0.253
Previous sports injuries (%)	322 (75.6%)	184 (76.0%)	506 (75.7%)	0.897
Current running injury (%)	405 (96.4%)	232 (97.5%)	637 (96.8%)	0.461
Attributed to Taroko Marathon (%)	271 (69.0%)	150 (65.8%)	421(67.8%)	0.416
Marathon experience (years)	2.22 ± 1.8	1.85 ± 1.65	2.09 ± 1.76	0.010*
Number of participations in marathons (times/year)	5.8 ± 13.01	4.28 ± 6.76	5.26 ± 11.24	0.140
Normal running (times/week)	2.73 ± 2.01	2.33 ± 1.71	2.59 ± 1.92	0.013 *
Weekly running distance (km)	8.46 ± 11.39	6.33 ± 3.3	7.75 ± 9.54	0.021 *
Weekly running time (min)	50.02 ± 25.45	43.17 ± 23.46	47.68 ± 24.96	0.026 *
Mountain marathon times	5.16 ± 12.06	3.63 ± 11.69	4.69 ± 11.95	0.362
Number of self-training sessions (times/week)	3.17 ± 2.3	2.32 ± 1.56	2.94 ± 2.15	0.061
Self-training distance (km)	11.89 ± 21.91	10 ± 5.32	11.41 ± 19.11	0.659
Warmup time (min)	14.75 ± 14.97	13.18 ± 12.26	14.19 ± 14.07	0.168
Need to warm up more	12.77 ± 9.5	11.9 ± 8.66	12.49 ± 9.22	0.546
Satisfaction of service provided by physical therapy	4.88 ± 0.42	4.87 ± 0.43	4.88 ± 0.42	0.774
VAS (Pre)	5.87 ± 1.76	5.76 ± 1.91	5.83 ± 1.82	0.459
VAS (Post)	2.28 ± 1.55	2.29 ± 1.57	2.28 ± 1.55	0.929

BMI = body mass index; VAS = visual analog scales; Pre = previous; Post = posterior. Data are presented as n or mean ± standard deviation. * *p* value < 0.05 was considered statistically significant after test.

**Table 2 ijerph-17-08101-t002:** Factors associated with injury (calf, thigh, and knee) in the lower extremities after the gorge marathon from 2013 to 2018 (n = 709).

Characteristics	Calf		Thigh		Knee	
	Odds Ratio (95% CI)	*p* Value	Odds Ratio (95% CI)	*p* Value	Odds Ratio (95% CI)	*p* Value
**Age**	1.00 (0.98,1.02)	0.704	0.98 (0.96,1.00)	0.103	0.98 (0.96,0.99)	0.019 *
**Sex**	-	-	-	-	-	-
Female	References	NA	References	NA	References	NA
Male	1.31 (0.86,2.00)	0.208	2.42 (1.40,4.19)	0.002 *	0.78 (0.51,1.19)	0.241
**BMI Group**	-	-	-	-	-	-
Normal	References	NA	References	NA	References	NA
Underweight	1.66 (0.76,3.61)	0.203	3.35 (1.41,8.01)	0.006 *	0.32 (0.12,0.81)	0.017 *
Overweight	1.00 (0.63,1.60)	0.993	1.16 (0.67,1.99)	0.593	1.11 (0.69,1.79)	0.666
Obese	1.79 (0.92,3.49)	0.088	1.49 (0.70,3.15)	0.303	1.25 (0.61,2.56)	0.540
**Education**	-	-	-	-	-	-
high school degree or below	References	NA	References	NA	References	NA
College degree	1.05 (0.51,2.17)	0.888	0.91 (0.38,2.14)	0.824	0.79 (0.37,1.70)	0.553
Master’s degree or above	0.98 (0.47,2.06)	0.967	1.16 (0.49,2.75)	0.743	0.91 (0.42,1.97)	0.803
**Type**	-	-	-	-	-	-
Mini-marathon	References	NA	References	NA	References	NA
Half-marathon	1.22 (0.52,2.87)	0.653	1.14 (0.39,3.33)	0.817	1.11 (0.46,2.66)	0.815
Marathon	0.91 (0.34,2.44)	0.843	1.56 (0.47,5.15)	0.466	0.92 (0.34,2.53)	0.874
**Previous sports injuries**	-	-	-	-	-	-
No	References	NA	References	NA	References	NA
Yes	4.02 (2.44,6.60)	<0.001 *	9.66 (4.84,19.27)	<0.001 *	5.65 (3.88,8.20)	<0.001 *

Data are presented as odds ratio (95% CI). * *p* value < 0.05 was considered statistically significant.

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
