# Peer review of "Common Running Musculoskeletal Injuries and Associated Factors among Recreational Gorge Marathon Runners: An Investigation from 2013 to 2018 Taroko Gorge Marathons"

_ijerph, 2020, doi:10.3390/ijerph17218101_

Round 1
Reviewer 1 Report
Thank you for letting me review the interesting manuscript entitle "Common running musculoskeletal injuries among recreational gorge marathon runners: an investigation from 2013 to 2018.
Overall, it is an interesting study with a good literature review and the results are easy to read and interpret. Congratulations.
However, I have few concerns in the Material and Methods section that should be explained and answered:
- The inclusion and exclusion criteria needs further explanation. The information provided in this section is too scarce.
- Concerning the questionnaire, was the questionnaire specially made for this study? In this case, was the questionnaire previously validated? If the questionnaire was not validated, it is a great limitation of the study and should be mentioned
- A detailed description of each part of the questionnaire may clarify the section. Each item of the questionnaire, what moment of the training or race did it refer to?
- Based on the information in the introduction, a running-related injury is a musculoskeletal disease caused by running, which results in a limitation for at least 1 week. When was the questionnaire filled in to consider this fact?
- The authors described open-ended question. How were those responses summarized and classified?
In the discussion part, there is explanation about the limitations of the study. This study present some limitations that should be pointed out at the end of the discussion part.
Finally, in my opinion, the conclusion is too long and should refer only to the most relevant part of the study. Rewrite the conclusion.
The inclusion of the questionnaire in an appendix would be clarifying for the potential readers and would allow the replication of the study and the comparison of results.
Author Response
"Please see the attachment".

Reviewer 2 Report
This manuscript is entitled "Common running musculoskeletal injuries among recreational gorge marathon runners: an investigation from 2013 to 2018" which aimed to explore the incidence of musculoskeletal injuries occurring in participants in the 2013 to 2018 Taroko Gorge Marathon and the distribution of running injuries and related influencing factors.
This study is quite interesting and present relevant data. However, some issues should be addressed:
1. Title
- My suggestion for title: "Common running musculoskeletal injuries and associated factors among recreational runners: an investigation from 2013 to 2018 Taroko Gorge Marathon". This is just a suggestion, but the bold information should be included in the title.
2. Abstract
- Information about the sample size and sample characteristics should be included.
- Include the country from this Marathon.
- Instruments and variables are not correctly presented.
- Statistical analysis should be included.
- You don't have any hypothesis in the abstracts, so "We confirmed" needs to be replaces by other words, for instance "We concluded..."
3. Introduction
- This section should be updated with recent references. Most of references are from more than 5 or 10 years ago.
- This sentence is confuse and should be rewritten " It is generally believed that running on 55 terrain and slopes may be related to the occurrence of injuries". Please, also cite some references that support this belief.
- As the Taroko Marathon is specific for a country, the authors should include more information about how important is this Marathon. How many participants per year? What country? Is the most important from the country?
- In the last paragraph, please, include the study hyphothesis.
4. Method
- Include the country.
- Include a subsection named "Marathon characteristics" and describe the Marathon. Short history, way, and others relevant aspects for the reader...
- Is the questionnaire valid and reproducible? What is the score of both? This should be included in the 2.2 section.
- The following sentence is confuse and there is a lack of concordance: " Risk factors related to injury included age, sex, BMI, education level, type of competition, and previous injuries". Please, rewrite it.
- How do you achieve the OR (odds ratio)? This is not mentioned in the method section. It is important state clearly that you performed the logistic regression and the effect measure is the OR.
5. Results
- Tables should be revised. Acronyms should be described in the legend. Also the meaning of bold numbers, etc.
- All figures should be revised. Please, remove horizontal lines inside the graph. Where is the vertical axis?
6. Discussion
- Limitation as well as perspectives should be included.
- In the Journal IJERPH there are several articles about running and exercise quite recent and should be read and cited.
Author Response
"Please see the attachment."

Round 2
Reviewer 1 Report
the authors have answered my questions and modified my suggestions. Congratulations for the work.
Reviewer 2 Report
All my comments were addressed.